# BOAS in the Boston Terrier: A healthier screw-tailed breed?

Francesca Tomlinson[1], Ella O'Neill[2], Nai-Chieh Liu[3], David R. Sargan[1], Jane F. Ladlow[1] *

1 Department of Veterinary Medicine, University of Cambridge, Cambridge, United Kingdom, 2 Lung Function Unit, Cambridge University Hospitals NHS Foundation Trust, Cambridge, United Kingdom, 3 School of Veterinary Medicine, Institute of Veterinary Clinical Science, National Taiwan University, Taipei, Taiwan

* jfl1001@cam.ac.uk

**Data Availability Statement:** All relevant data are within the manuscript and its Supporting Information files.

## Abstract

Brachycephalic obstructive airway syndrome (BOAS) is well documented in the three most popular brachycephalic dog breeds of the UK and several other countries: French Bulldogs, Pugs and Bulldogs. More extreme conformation has been found to be associated with increased risk of BOAS and other brachycephalic disease in these breeds, such as ocular, neurological, and dental disease. Less is known about how BOAS and other brachycephalic conformation-related disease affects other breeds such as the Boston Terrier. In this study, one-hundred and seven Boston Terriers were prospectively recruited from the UK dog population and underwent clinical assessment, respiratory function grading and conformational measurements. Whole-body barometric plethysmography was used in a smaller cohort of dogs to compare the quantitative differences in respiratory parameters between both affected and unaffected Boston Terriers, and control mesocephalic dogs. When compared to an equivalent study population of French Bulldogs and Bulldogs, it was found that Boston Terriers have a significantly higher proportion of BOAS Grade 0 dogs at 37.5% compared to 10% and 15.2% respectively (p<0.01). Within the breed, more extreme brachycephalic conformation was found to be associated with an increased risk of BOAS: specifically, nostril stenosis, facial foreshortening, abnormal scleral show, and higher neck to chest girth ratio. However, there is considerable overlap between measurements of affected and unaffected dogs in these variables. Therefore, the use of respiratory function grading is likely to be more advantageous for owners, breeders, and veterinary surgeons in accurately selecting unaffected dogs.

## Introduction

Brachycephalic obstructive airway syndrome (BOAS) describes a number of clinical signs caused by upper airway obstruction in brachycephalic dog breeds. It is primarily described in three popular extreme brachycephalic breeds: the French Bulldog, the Pug, and the Bulldog [1, 2]. Abnormal breathing sounds, stertor and stridor, are signifiers of the disease, and dogs often show increased respiratory difficulty under exercise, stress, and heat [3, 4]. BOAS can

**Funding:** This work was supported by a grant from the Kennel Club Charitable Trust. The funders had no role in study design, data collection and analysis, decision to publish, or preparation of the manuscript.

**Competing interests:** The authors have declared that no competing interests exist.

vary substantially in severity, with the worst dogs affected at risk of collapse and death due to acute airway obstruction [5]. Affected dogs can also suffer from sleep disorders [6, 7], recurrent gastro-oesophageal reflux [8, 9] and a state of chronic hypoxia which can lead to wide-ranging systemic effects [10, 11].

An increase in puppy registrations of these three breeds in recent years has led to concern over a looming welfare crisis [12], particularly given reported under-recognition and normalisation of the disease [13]. Many campaigns have been launched to raise awareness and education about the potential issues of these breeds and a UK Brachycephalic Working Group was formed to bring together key stakeholders [14]. Different approaches have been undertaken by various countries and organisations to mitigate the impacts on animal welfare. The Netherlands has introduced a ban affecting a number of brachycephalic breeds based on conformation criteria [15], whilst in Norway legal cases have been brought against specific breeds [16]. In the UK, a Respiratory Function Grading scheme was launched by the Kennel Club and University of Cambridge [17] and this has subsequently been adopted by several other countries. However, this is currently limited to French Bulldogs, Pugs and Bulldogs.

The Boston Terrier is a brachycephalic dog breed derived from the crossing of the Bulldog and terriers originating in Boston, United States during the 19th century. In the year 2022, 1,603 Boston Terriers were registered with the UK Kennel Club compared to 11,667 Bulldogs and 42,538 French Bulldogs [18]. Like the French Bulldog and Bulldog, the Boston Terrier is considered one of the 'screw-tailed' dog breeds that share common ancestors [19], and whereby the trait of having a short, kinked tail is ubiquitous within the breed. Recent genetic studies identified a frameshift mutation in the *DISHEVELLED 2* (*DVL2*) gene which is proposed to contribute to a Robinow-like syndrome in the screw-tailed breeds [20]. The variant was found to be fixed in the Bulldog and French Bulldog, and a high allele frequency (0.94) in the Boston Terrier. In another study, the variant gene was homozygous in all one-hundred and sixty-five Boston Terriers tested [21]. Alongside BOAS, the brachycephalic and screw-tailed phenotypes have been associated with a number of conditions including vertebral malformations [21, 22], ocular disease [23, 24], skin fold dermatitis [25], dental abnormalities [26–28] and dystocia [29].

Whilst the extent of BOAS and other brachycephalic-related disease has been well documented in French Bulldogs, Pugs and Bulldogs [30–32] the extent to which Boston Terriers suffer from comparable brachycephalic-related health issues is less clear. The Boston Terrier is less numerous in the UK than the other breeds, therefore the evidence base is not as well established. There are key differences in the most prevalent anatomical lesion sites of BOAS across the three most popular breeds. For example, Bulldogs are predisposed to tracheal hypoplasia whilst Pugs are more prone to a higher grade of laryngeal collapse [33, 34]. However, evidence as to how Boston Terriers are affected is limited. Boston Terriers have been reported as undergoing BOAS treatment or investigation in the veterinary literature [35–37]. Although in these studies, Boston Terriers are examined in smaller numbers. Possible explanations for a smaller sample population in these studies are that fewer Boston Terriers are reported due to their comparative rarity within the wider dog population. It also remains possible that they do not suffer as overtly with BOAS as the other breeds and proportionally less individuals are clinically affected.

This prospective study aimed to investigate the prevalence of BOAS in Boston Terriers and identify risk factors associated with the disease within the breed. The study hypothesised that Boston Terriers are less affected by BOAS than the other two screw-tailed breeds, and this was tested through the use of respiratory function grading and mid-expiratory flow analysis using whole-body barometric plethysmography (WBBP). The study evaluates to what extent external conformational factors associated with BOAS that have previously been identified in other

breeds are applicable in Boston Terriers [38, 39]. This could be useful to assist breeders and owners in selecting dogs that are most likely to be at lower risk of disease. Understanding how Boston Terriers are affected by BOAS and why this may differ from the other breeds could offer insight for comparative studies into the key factors which drive the pathophysiology of the disease.

## Methodology

### Subjects

Boston Terriers were prospectively recruited between September 2021 to August 2023 from the pet dog, breeding and showing population (n = 107). Some dogs attended individual appointments for participation at the Queen's Veterinary School Hospital (QVSH) in Cambridge (n = 34), whilst others were recruited at dog shows or breed-specific health testing days (n = 73). Dogs included in the study were over 12 months of age and were identifiable as Boston Terriers. Dogs reported by the owner to have had previous airway surgery (n = 3) were excluded from the BOAS assessment data. Study information was distributed to the owners and the study was performed under ethical approvals CR530 and CR562 from the Department of Veterinary Medicine, University of Cambridge. Owners were given study information and written informed consent was obtained for owners of participants documented by the signing of a consent form.

### Clinical assessment

A clinical examination was performed in all subjects (n = 107) to assess the overall health of the dog prior to respiratory testing, with particular reference to known brachycephalic health issues. Thoracic auscultation was performed and if any clinical signs or history of lower airway disease was apparent, the subject was excluded from the study (n = 0). A partial neurological examination was performed using the validated Texas Spinal Cord Injury Scale (TSCIS) [40] to test proprioception, spinal reflexes and assess gait. Any skin lesions were rated using the Canine Allergic Dermatitis Extent and Severity Index (CADESI)-4 [41]. Skin folds were recorded and the extent of intertrigo rated from none (defined as inflammation or alopecia absent), mild (alopecia and/or erythema present with discharge absent), moderate (erythema, alopecia, and discharge present) or severe (self-trauma or ulceration present). Ocular lesions such as discharge, ectropion, entropion, and abnormal scleral show were recorded by quadrant if present. Dental abnormalities observable on conscious examination, such as malocclusions, mandibular prognathia, retained deciduous teeth were recorded and periodontal disease staging noted from a scale of 0-IV [42].

### Respiratory function grading

Respiratory function grading was performed in dogs that met the inclusion criteria (n = 104) using the established methodology previously validated with whole-body barometric plethysmography [4, 17]. Following thoracic auscultation, pharyngo-laryngeal auscultation was performed to assess for noises indicative of upper airway obstruction before and after an exercise test whereby the dog is run at approximately 3–5 miles per hour for 3 minutes. Stertor (low-pitched sounds) and stridor (high-pitched sounds) were graded as none, mild, moderate, or severe. If a dog displayed nasal stridor, rather than laryngeal stridor, this was recorded in the study notes. The dog was also observed closely for additional signs of breathing difficulty including abdominal effort or cyanosis. Dogs that were able to complete the exercise test with no dyspnoea and no stertor or stridor detected were classified as Grade 0 (as under the existing

respiratory function grading scheme). Dogs that displayed any upper airway noises were classified as from Grades 1 to 3. Grade 1 indicates mild BOAS with no exercise intolerance or signs of being clinically affected. Grade 2 indicates moderate BOAS that is clinically significant, whilst Grade 3 is severe BOAS that results in an inability to exercise or signs of respiratory distress, such as cyanosis. Nostril stenosis was graded according to existing published criteria [38].

## Whole-body barometric plethysmography

Whole-body barometric plethysmography (WBBP) has been used to assess differences in respiratory parameters in unrestrained, conscious dogs affected by BOAS [1, 43]. This can be used as a non-invasive respiratory function test. Limitation in expiratory flow rates at the time-points 0.25, 0.5 and 0.75 of the expiratory breath have been used to detect obstruction in human respiratory medicine [44–47]. In this study, WBBP was used to quantify the difference in mid-expiratory flow rate change between BOAS Grade 0 and BOAS Grade 1 to 3 Boston Terriers, and also compare to control mesocephalic dogs.

A smaller cohort of Boston Terriers in this prospective study underwent WBBP (n = 26), with fourteen Boston Terriers classified as BOAS Grade 0 (n = 14) and twelve classified as BOAS Grade 1–3 (n = 12) following respiratory function assessment. Control dog data (n = 10) was collected from a number of mesocephalic breeds (Miniature Schnauzer (n = 2), Labrador (n = 1), Jack Russell Terrier (n = 2), Hungarian Vizsla (n = 2), Springer Spaniel (n = 1), crossbreed (n = 1)). The methodology used in this study for collection of respiratory traces follows the same protocol previously described [1, 43]. The WBBP chamber used was the model ElectroMedical Measurement Systems (EMMS) Model PLY-361 for dogs. Transducer signals were amplified using a strain gauge amplifier and translated to commercial software ESS-102 EMMS Data Acquisition, eDacq. Respiratory traces were collected from a period of approximately 20 minutes which includes video recording of the dog to enable the selection of breaths at a normal respiratory rate (15–40 bpm).

Representative individual breaths were selected from the traces by a clinical respiratory physiologist (EO). A total of five to ten breaths were selected when the patient was in a single position and the ventilation was relaxed tidal breathing. The inclusion and exclusion criteria for accepted breaths are detailed in Table 1. Mid-expiratory flow (MEF) rates at each quartile of the expiratory breath (MEF 0.25, 0.50 & 0.75) were recorded and the mean of the selected breaths was calculated for each dog. The mean MEFs were then normalised to body weight to account for differences in size and lung capacity between dogs. The mean body weights of the sample populations were 14.1kg (standard deviation (SD): 8.7) for control dogs, 8.3kg (SD: 1.8) for unaffected Boston Terriers and 8.8kg (SD: 3.0) for BOAS affected Boston Terrier.

## Conformational measurements

**Soft tape measurements.** Soft tape measurements were used to collect the five body measurements to describe conformation as described in Table 2 and illustrated in Fig 1: body

**Table 1. Inclusion and exclusion criteria for breaths accepted in the mid-expiratory flow rate analysis.**

| Inclusion | Exclusion |
|---|---|
| Tidal Breathing | Panting, coughing, yawning or straining when breathing |
| Clean recording | Artefact |
| Patient in relaxed position, sitting or lying | Moving, stretching, or lying upward |
| At least 2 breaths continually matching in appearance | Single breaths between movement or change in breathing type |

**Table 2. Conformational measurements taken through soft tape and photographic measurements.**

| Measurement | Method of Collection | Anatomical Landmarks |
|---|---|---|
| **Body length (BL)** | Soft tape | Distance from cranial edge of scapula to tail root on the dorsal midline. |
| **Body height (BH)** | Soft tape | Distance from cranial edge of scapula (on midline) down forelimb to level of floor. |
| **Neck girth (NG)** | Soft tape | Mid-circumferential girth of neck. |
| **Chest girth (CG)** | Soft tape | Girth measured at deepest level of chest. |
| **Tail length (TL)** | Soft tape | Distance from root of the tail to the most caudal edge. |
| **Skull width (SW)** | Frontal plane and dorsal plane photographs | Linear distance between the most lateral points of the zygomatic arches. |
| **Intercanthal distance (ICD)** | Frontal plane photographs | Linear distance between the medial canthi. |
| **Skull length (SL)** | Dorsal plane photographs | Linear distance from occiput to rostral edge of nasal planum. |
| **Muzzle length (MzL)** | Sagittal plane photographs | Linear distance from the stop to the rostral edge of nasal planum. |
| **Cranial length (CrL)** | Sagittal plane photographs | Circumferential distance from the stop to the occiput in midline sagittal plane. |

length (BL), body height (BH), neck girth (NG), chest girth (CG) and tail length (TL). Body length, neck girth and chest girth have been described in previous studies investigating conformation in brachycephalic dogs [38, 39]. Neck to chest girth ratio (NGR) was subsequently calculated by dividing neck girth by chest girth (NGR = NG/CG).

**Photographic measurements.** Five measurements were taken from photographs of study dogs to describe brachycephaly and facial conformation, as detailed in Table 1 (intercanthal distance (ICD), skull width (SW), skull length (SL), muzzle length (MzL) and cranial length (CrL)). This was done using the computer software Inkscape 2020 (Inkscape, Available at: https://inkscape.org). Photographs were taken in three views to capture the three-dimensional parameters of the subject's craniofacial morphology, as described in Fig 2. Three photographs

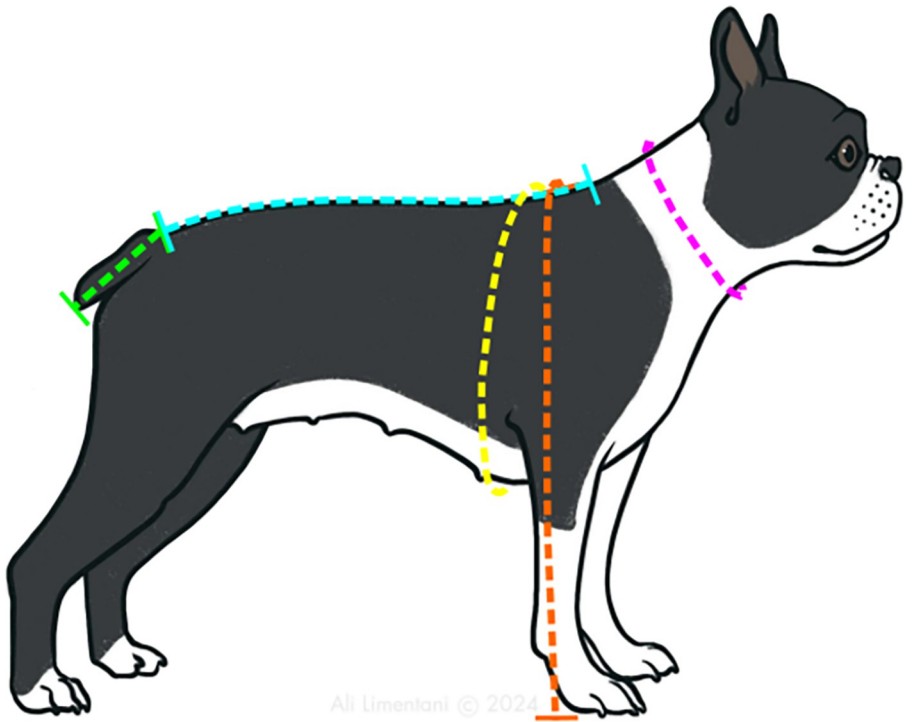

**Fig 1. Soft tape body measurements.** Line colours corresponding to the following measurements: (a) Blue: body length, (b) Orange: body height, (c) Pink: neck girth, (d) Yellow: chest girth, (e) Green: tail length.

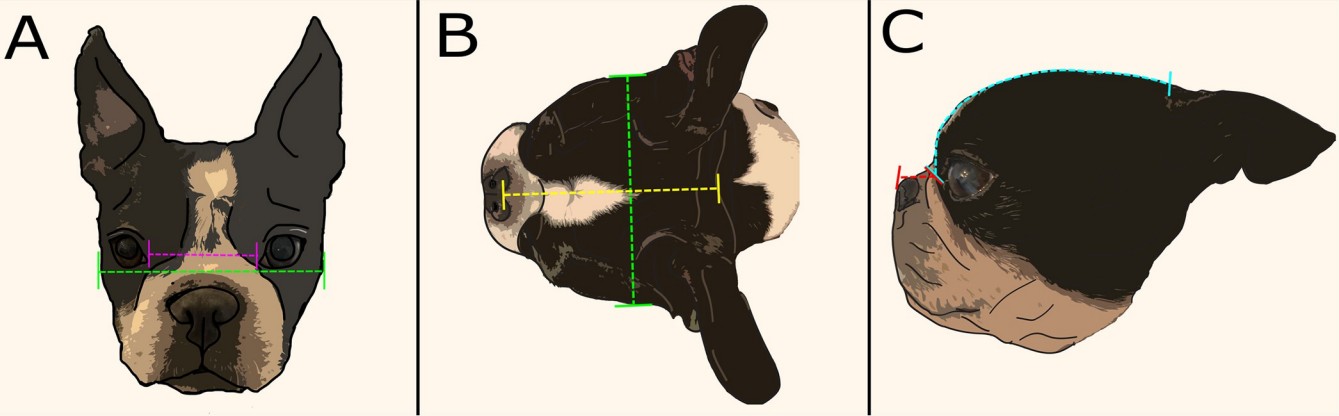

**Fig 2. Photographic conformational measurements.** (A) Frontal plane (FP) photographs were taken with the camera facing directly from rostral to caudal, with the subject looking directly at the camera lens. Measurements include ICD (pink line) and SW (green line). (B) Dorsal plane (DP) photographs were taken with the camera looking down at the subject from a bird's eye view; dorsally to ventrally and perpendicular to the frontal plane. Measurements include SW (green line) and SL (yellow). (C) Sagittal plane (SP) photographs were taken as a side-on view, directly lateral to the subject and parallel to the midline sagittal plane. Measurements include MzL (red line) and CrL (blue line).

of each subject were taken in each plane, providing nine photographs for measurement, thereby producing three repeated measurements from which the mean was calculated. For inclusion, photographs were required to be visible in the correct plane without distortion and have no excessive fur or hair impeding view of the anatomical landmarks for measurement.

Three ratios were calculated from these measurements. Eye width ratio (EWR) was calculated from the frontal plane measurements as EWR = ICD/SW (43). Skull index (SI) was calculated using photographic measurements from the dorsal plane measurements as skull width (SW) divided by skull length (SL): SI = SW/SL (48). Craniofacial ratio (CFR) was calculated as muzzle length divided by cranial length: CFR = MzL/CrL [39].

## Statistical methods

**BOAS grade distribution.** Previously published data available on the grade distribution in a similar study on French Bulldogs, Bulldogs and Pugs was used as a comparison to the percentage of BOAS affected dogs in this study [1]. The proportion of Grade 0 dogs within the sample population was compared to the proportion of Grade 0 Boston Terriers within this current study using a one-sided Fisher's test.

**Mid-expiratory flow rate analysis.** Means, standard deviations and 95% confidence intervals were calculated for the change in MEF at three time periods (0.25 to 0.75, 0.25 to 0.5 and 0.5 to 0.75) within each group and flow rate changes were compared using paired two-tailed t-tests. Comparison of the change in MEF rates between the different groups was analysed by repeated measures ANOVA using the mixed-model approach in which subjects are assigned as random effects. The mixed model uses a compound symmetry covariance matrix and is fit using the Restricted Maximum Likelihood (REML) approach. In the absence of missing values, P values and multiple comparison tests given by this method are the same as would be obtained by repeated measures ANOVA. Unpaired two-tailed t-tests were used to look at the differences in flow rate change dynamics between the groups within the three time periods of the breath. The P values obtained from the multiple t-test comparisons were adjusted using the Holm-Sidak method.

**BOAS risk factor analysis.** All photographic measurements used in the analysis for BOAS risk factors were undertaken by the same investigator (FT). Univariate analysis was

initially performed for the continuous conformational measurements, both soft tape and photographic to compare Grade 0 Boston Terriers to Grades 1–3. A one-tailed student's t test (unpaired, parametric) was used for most continuous variables (BL, BH, NGR, TL, age, and weight) and the F test performed to test for equality of variance. Welch's correction was applied to the variables that did not demonstrate equality of variance (EWR, SI, CFR). For the categorical variable data (nostril stenosis, facial fold, scleral show, BCS, sex and neuter status), Fisher's exact test was performed. Multiple logistic regression analysis was performed using the conformation or signalment variables that produced a p value less than 0.1 in the univariate analysis indicating statistical significance or nearing statistical significance for association with BOAS status (Grade 0 versus Grade 1–3) [48]. Classification cut-off was set at 0.5. Categorical variables were transformed (abnormal scleral show: 0 = absent, 1 = present. Nostril stenosis: 0 = open, 1 = mild stenosis, 2 = moderate stenosis, 3 = severe stenosis) and ratios were increased by a factor of 100 (CFR, NGR & SI).

**Repeatability of conformational measurements.** The intra- and inter-observer repeatability of conformational measurements was assessed using intra-class correlation coefficient. Bland-Altman analysis was used to compare photographic CFR measurements with equivalent computed tomography (CT) measurements. Full methodology and results can be found in supporting information file S1 Appendix, S1 Table and S1 Fig.

**Statistical software.** Calculation of intraclass correlation coefficient used to validate conformation measurements analysis was undertaken using R Statistical Software [49]. All other statistical analysis was undertaken using GraphPad Prism version 9.5.0 for Mac OS, GraphPad Software, San Diego, California USA, www.graphpad.com. In this paper, results described as significant indicate a p value of p<0.05. Results between p = 0.05 and p = 0.1 were considered nearing significance, and results of p<0.01 were suggestive of strong support of the alternative hypothesis [48].

## Results

### Sample population characteristics

A total of 104 Boston Terriers took part in the study and underwent full BOAS assessment. Eighty-nine (90%) Boston Terriers were Kennel Club (KC) registered; registration status was unknown in five dogs. Sixty-six dogs (64%) were female and 38 (36%) were male. Sixty-five were intact and 35 dogs were neutered. Median age was three years (range: 1–10) and median body condition score was 5/9 with a mean weight of 7.8kg (range: 4.2–15.8kg). No significant differences were found between the Grade 0 and Grade 1–3 sample populations for sex (p = 0.83) or neutering status (p = 0.14). Median age was slightly higher in the Grade 1–3 Boston Terriers (3 years compared to 2.5 years); however, this was not found to be statistically significant (p = 0.18).

### Proportion of BOAS affected dogs

A total of 39/104 (37.5%) Boston Terriers had no observable upper respiratory noise nor excessive respiratory effort on respiratory function testing, thereby were assigned as Grade 0. Compared to previously published data on French Bulldogs and English Bulldogs in a comparable study population [1] (Fig 3), Boston Terriers have a higher proportion of dogs being Grade 0 than Bulldogs (15.2%, p = 0.001) and French Bulldogs (10%, p<0.001). The second most prevalent group in Boston Terriers was Grade 1 (n = 37, 35.6%) followed by Grade 2 (n = 27, 26.0%). Only one Boston Terrier was assigned a Grade 3 (n = 1, 0.96%).

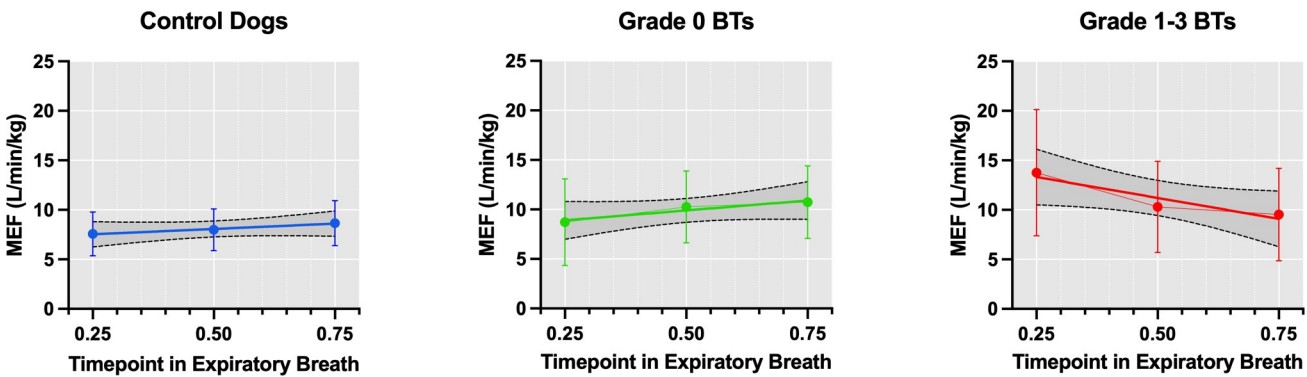

## BOAS Grade Distribution

**Fig 3. The BOAS grade distribution of Boston Terriers in the sample population.** Percentages compared to that of graded of Bulldogs and French Bulldogs in the previous study by Liu et al. [1].

### Whole-body barometric plethysmography findings

The expiratory flow traces of Grade 1–3 compared to Grade 0 Boston Terriers and control dogs were found to have different shapes, as demonstrated in Fig 4 whereby mean mid-expiratory flow (MEF) rates are plotted across three timepoints in expiration. Fig 5 illustrates

**Fig 4. Mean mid-expiratory flow rates recorded at each quartile of the expiratory breath.** Considered for three sample populations: 1. Control dogs, 2. BOAS Grade 0 Boston Terriers (BTs) and 3. BOAS Grade 1–3 Boston Terriers (BTs). Flow rates have been normalised to body weight (kg). Error bars displayed and 95% confidence intervals represented by dashed lines.

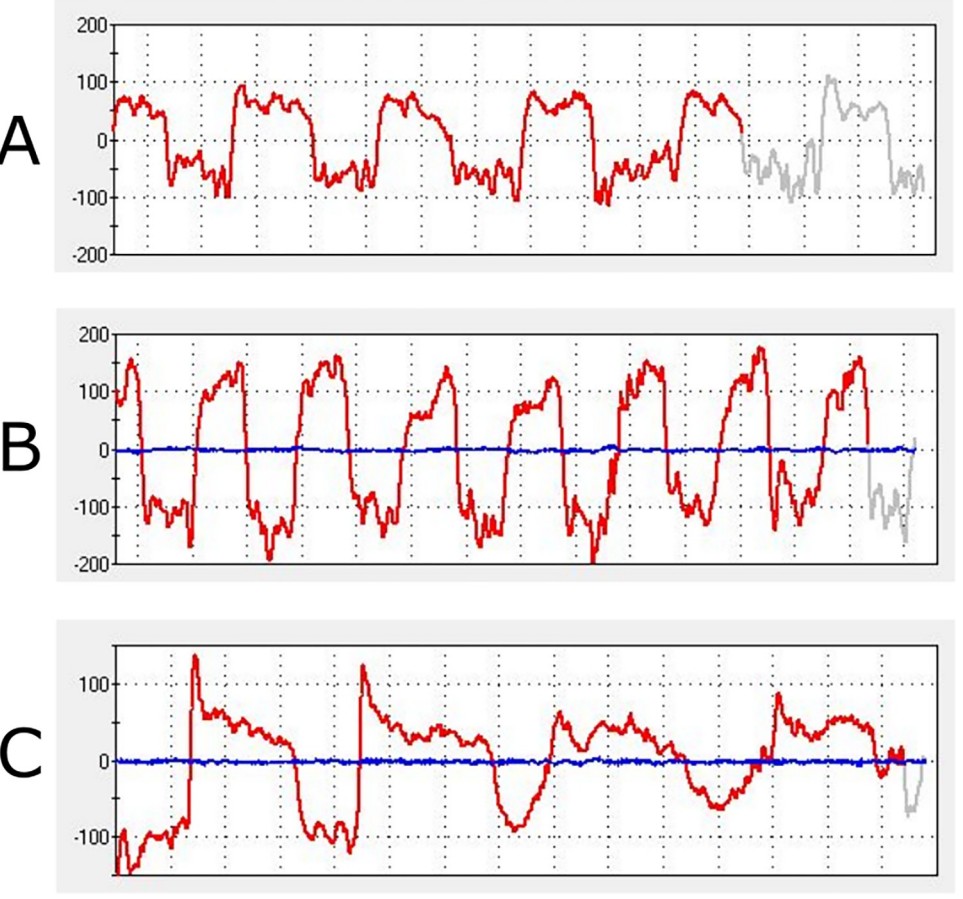

**Fig 5. Example whole-body barometric respiratory flow traces from the three sample populations.** (A) Control dog. (B) Grade 0 Boston Terrier. (C) Grade 2 Boston Terrier.

examples of respiratory flow traces from the three sample groups. The Grade 1–3 Boston Terriers had higher mid expiratory flow (MEF) rates at the first quartile of the expiratory breath, which reduced over the breath with a mean reduction in flow rate of -4.23L/min/kg (raw p = 0.038, adjusted p = 0.074). In contrast, the traces of the control dogs and Grade 0 Boston Terriers had a smaller change in flow rate over the expiratory breath with a mean increase of 1.08L/min/kg (raw p = 0.003, adjusted p = 0.01) and 2L/min/kg respectively (raw p = 0.042, adjusted p = 0.074). The mixed effects model (REML) demonstrated a significant change over time for the control and Grade 0 dogs (p = 0.008), but no significant difference between these two groups over time (p = 0.385). There was a significant difference in the MEF rates over time between the Grade 1–3 Boston Terriers and the control dogs (p = 0.008) and Grade 0 Boston Terriers (p = 0.0006). Full results from the analysis can be found in supplementary file S2 Table.

## Risk factors for BOAS

**Conformation.** In the study population, mean body length was 31cm (95% CI: 30–32), mean body height was 38cm (95% CI: 37–39) (Fig 6). The dogs in the study sample had a mean weight of 8.2kg (95% CI: 7.8–8.6). There was no significant difference between Grade 0 and Grade 1–3 Boston Terriers in body length (p = 0.37), height (p = 0.38), or weight

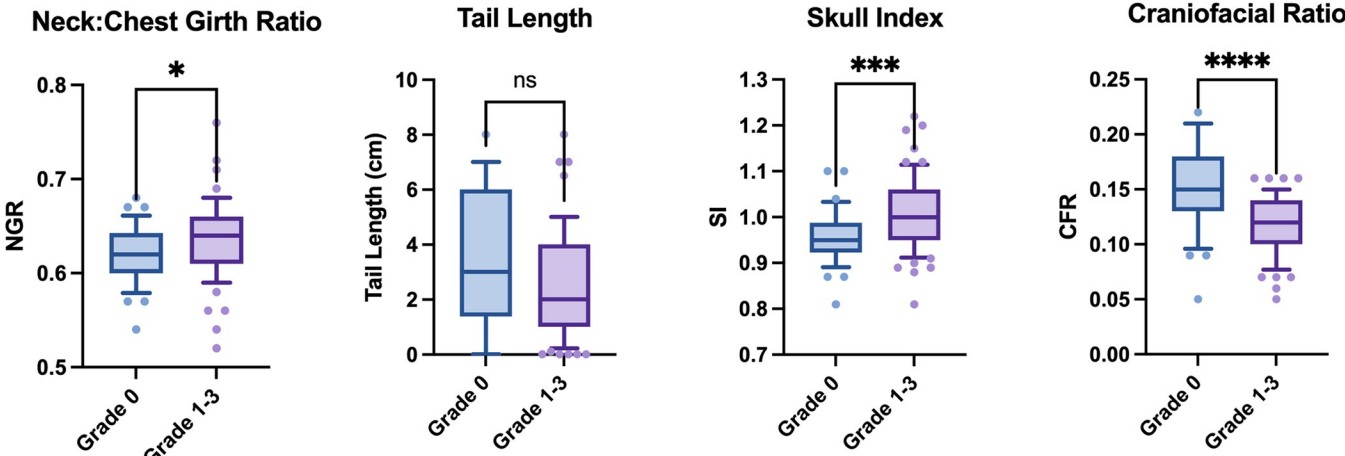

**Fig 6. Boxplots of conformational measurements in Grade 0 versus Grade 1–3 Boston Terriers.** Neck to chest girth ratio (p = 0.039), tail length (p = 0.052), skull index (p = 0.0002) and craniofacial ratio (p<0.0001).

(p = 0.88). The neck girth ratio was found to be significantly increased (p = 0.039) (in Grade 1–3 Boston Terriers (mean = 0.63, SD: 0.04) compared to Grade 0 Boston Terriers (mean = 0.62, SD: 0.03), indicating that more BOAS affected Boston Terriers had proportionately thicker necks than the Grade 0 Boston Terriers. Grade 1–3 Boston Terriers also had shorter tails than the Grade 0 participants with a mean of 2cm (SD: 2.0) compared to 3cm (SD: 2.4), however this was of weaker statistical significance (p = 0.05). In terms of the facial conformation measurements, there was no difference noted in the eye width ratio (p = 0.15), however strong evidence was found for the difference in both skull index and craniofacial ratio between the groups (SI: p = 0.002 & CFR: p<0.0001). A greater skull index was associated with being BOAS Grades 1–3, meaning that having a proportionately wider skull was associated with BOAS. A smaller craniofacial ratio was also associated with BOAS, indicating a greater risk for dogs with a proportionately shorter muzzle. Full descriptive statistics and unpaired t-test analysis results can be found in supplementary files S3 and S4 Tables.

Two categorisations of conformation were found to be significantly associated with BOAS status (Fig 7). Nostril stenosis was significantly associated with BOAS Grade 1–3 Boston Terriers (p = 0.002). Boston Terriers that had abnormal scleral show recorded were also significantly more likely to have BOAS (p = 0.04). Presence of a facial fold was not found to be associated with BOAS status (p = 0.19). Although a greater percentage of Grade 1–3 Boston Terriers were recorded as being overweight (19.2%, 95%CI: 11.8–29.7) compared to Grade 0 dogs (10.3%, 95%CI: 4.06–23.6), in this analysis, body condition score was not found to be significantly associated with BOAS status (p = 0.15).

Multiple logistic regression analysis was performed using the six conformation variables selected that had a p value less than 0.1 in the univariate analysis (nostril stenosis, scleral show, NGR, tail length, SI and CFR) (Fig 8). The area under the ROC curve was 0.868 (95%CI: 0.788–0.949, p<0.0001). The model with a negative predictive power of 80.6% and a positive predictive power of 83.9%. The model correctly classified 82.8% of subjects (73.5% BOAS Grade 0 and 88.7% BOAS Grade 1–3). The analysis indicated the two variables with the strongest evidence for association with BOAS status, were nostril stenosis (odds ratio (OR): 2.51, 95%CI: 1.18–5.85, p = 0.02) and craniofacial ratio (OR: 0.757, 95%CI: 0.603–0.919, p = 0.009) with being BOAS Grade 1–3, thereby indicating that dogs with more severe nostril stenosis

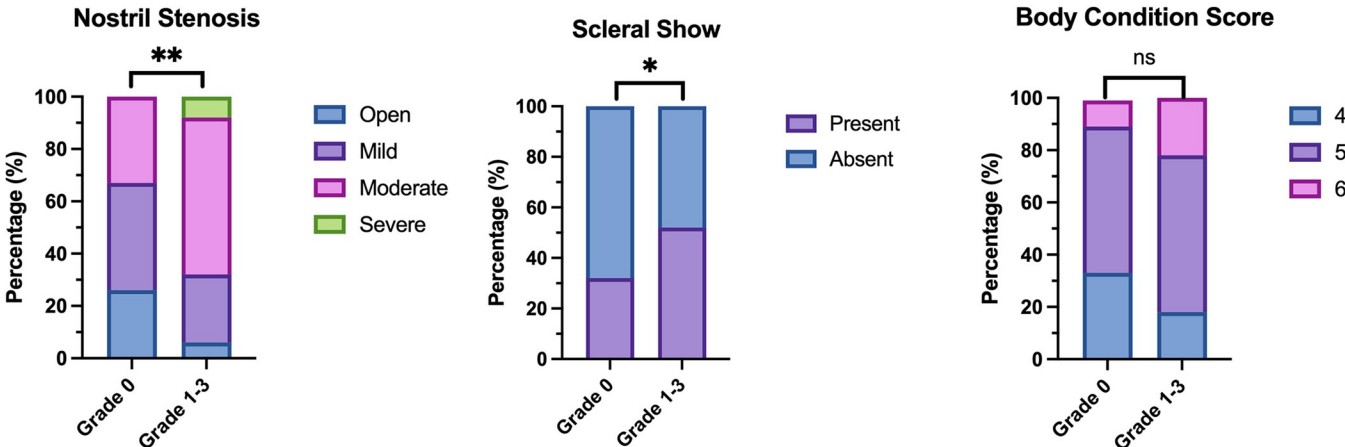

**Fig 7. The percentage of BOAS Grade 0 versus BOAS Grade 1–3 Boston Terriers across different subjective conformation variables.** Nostril stenosis (p = 0.002), scleral show (p = 0.04) and body condition score (p = 0.15).

and smaller craniofacial ratio are more at risk of being BOAS Grade 1–3. Full results of the analysis are detailed in S5 Table.

## Additional findings

One-hundred and seven (n = 107) Boston Terriers underwent full clinical examination. Mandibular prognathia was a ubiquitous finding in Boston Terriers. Facial skin folds were noted in 39 out of 93 dogs (41%). Of these, eight dogs had mild dermatitis and one had moderate dermatitis. A lip fold was noted in one dog and a tail fold noted in one other. The vast majority of Boston Terriers did not have any other skin lesions detected (89.5%). Out of the other 10.5%, six dogs (5.7%) were noted to have erythema, and additionally there were a number with lichenification (n = 1) and alopecia or excoriation (n = 4). Erythema was noted on the muzzle

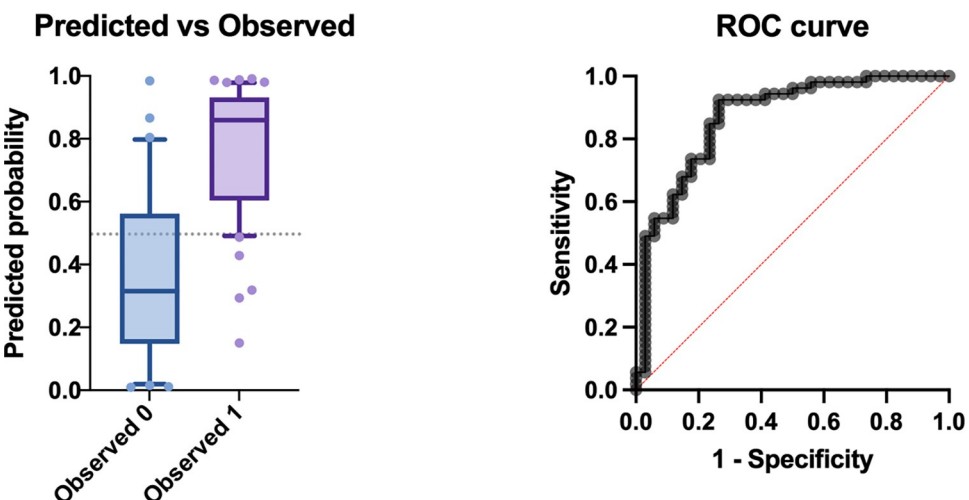

**Fig 8. Multiple logistic regression analysis of selected signalment and conformational factors.** Six variables included in analysis (nostril stenosis, scleral show, NGR, tail length, SI and CFR) with classification cut-off value at 0.5. Boxplot displaying the probability of the model predicting observed BOAS Grade 0 (0) and BOAS Grade 1–3 (1). Receiver operating characteristic (ROC) curve of the model with area under curve of 0.868 (Std. error: 0.0411, 95% CI: 0.788–0.949, p<0.0001).

(n = 3), ears (n = 2), axilla (n = 2), ventrum (n = 3) and feet (n = 2). External ear examination was unremarkable in 104 dogs, and abnormalities were noted in only three dogs; erythema (n = 2) and discharge (n = 1). Twenty-eight (26%) dogs had no ocular abnormalities detected. One dog had medial corneal pigmentation. Seventy-eight (74%) dogs were noted to have abnormal scleral show. Ocular discharge was noted in a small number of dogs (n = 8). No active cases of entropion, ectropion or nicitans gland prolapse were detected in the sample population. Out of 103 dogs, thoracic auscultation detected a heart murmur in five Boston terriers (4.9%) and was either Grade 1 (n = 3) or Grade 2 (n = 2). In all other dogs, no abnormalities were detected on thoracic auscultation. Sixteen dogs had patella luxation (unilateral n = 9, bilateral n = 7). On neurological examination, six dogs had deficits in proprioception shown by delayed hindlimb placement or hopping, or spinal reflex by delayed limb withdrawal. Phantom scratching was noted in one dog, and two dogs were reported to have been diagnosed with Chiari-like malformation or syringomyelia.

## Discussion

The findings from this study suggest that BOAS presents at a reduced prevalence in the Boston Terrier compared with the other two popular screw-tailed dog breeds: the French Bulldog and the Bulldog. Whilst there is still a substantial proportion of Boston Terriers that display some degree of respiratory noise, it appears that there are very few severe cases, compared to the most commonly affected breeds. This may be due to a smaller population given the rarity of the breed. However, there was only one dog described in this study that was assigned the most severe grade for BOAS (Grade 3).

A number of conformational risk factors for BOAS in the Boston Terrier have been highlighted in this study as illustrated in Fig 9: CFR, nostril stenosis, NGR, SI and scleral show. Nostril stenosis is the only visible upper airway obstruction site and is highly correlated with BOAS affected status in the Boston Terrier. The two measures of brachycephaly (SI & CFR) showed that more extreme facial foreshortening was correlated with increased risk of being BOAS affected.

Packer et al. [39] reported a comparable average craniofacial ratio for Boston Terriers (0.14), and this was reported as being less than that of the French Bulldog (0.19) and Bulldog (0.22). Despite an apparently lesser craniofacial ratio than the Bulldog and French Bulldog, the Boston Terriers appear less severely affected by BOAS. Neck girth ratio was found to be significantly associated with BOAS status in Boston Terriers, meaning that a thicker neck carries a greater risk of suffering from the disease. This is comparable to sleep apnoea in humans [50]. Neck girth ratio reported by Liu et al. [38] in Pugs (BOAS -ve 0.67, BOAS +ve 0.67) French Bulldogs (0.67, 0.70), and Bulldogs (0.66, 0.70), is greater than that of Boston Terriers (0.62, 0.63). In that study, NGR was found to have highly significant associations with BOAS in the bulldog breeds (French Bulldogs $p<0.0001$, Bulldogs $p<0.0001$), but not Pugs ($p>0.05$).

Any factors that increase strain on the respiratory system result in a worsening manifestation of BOAS and severe clinical signs such as cyanosis, collapse, and death. Pain, stress, and high environmental temperatures can all contribute to this. The overall shape and conformation of Boston Terriers is leaner in comparison to the heavily muscled Bulldog breeds or Pugs that are more prone to obesity [51]. In humans, a higher BMI can lead to reduced lung volumes compromising ventilation [52]. A higher body weight in relation to the height and size of the dog may result in an increased oxygen demand without having a more efficient airway to cope with it. The Boston Terrier is slighter and more athletic than the other Bulldog breeds, which could account for the better respiratory function and apparent adaptation to an increased degree of facial foreshortening.

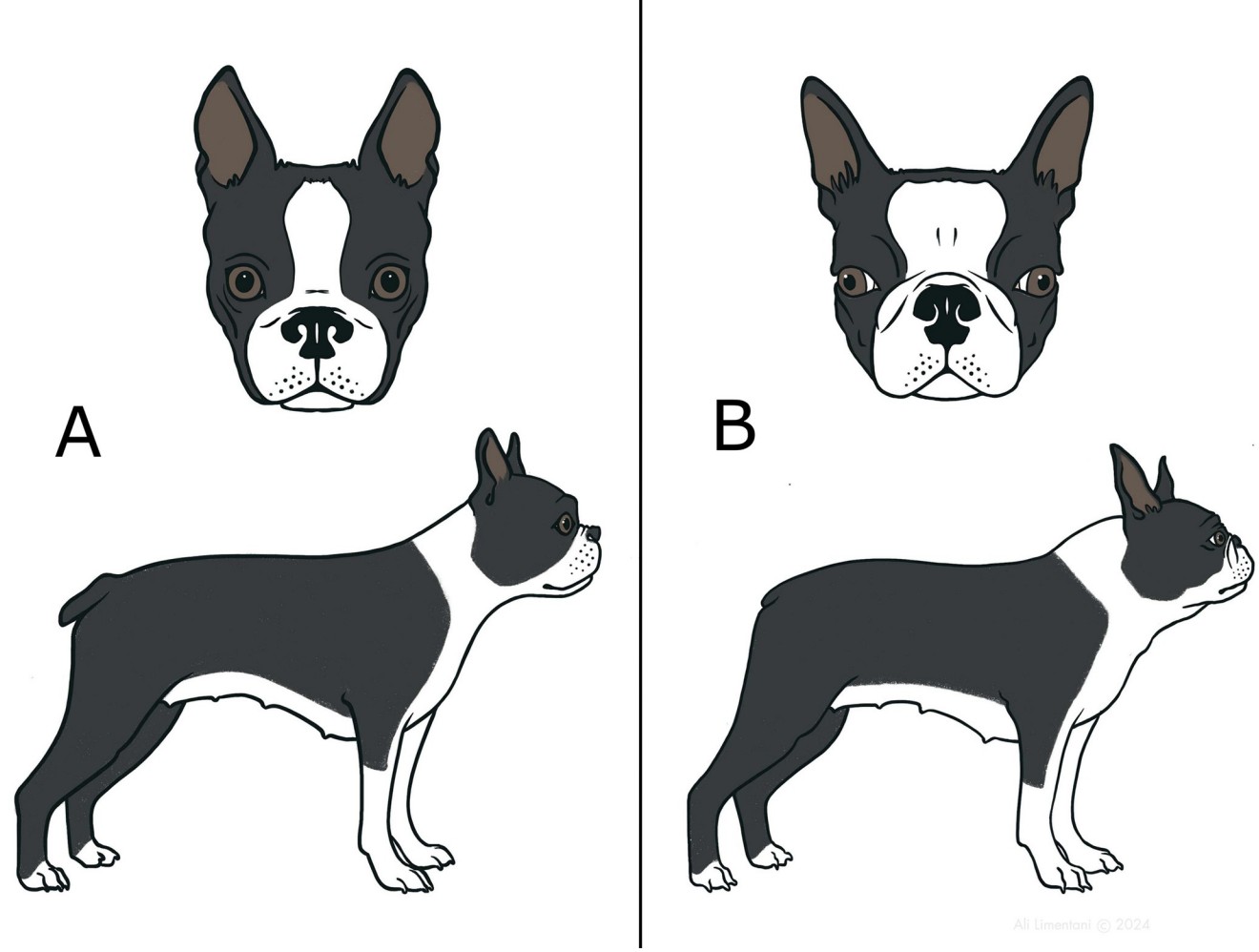

**Fig 9. Comparative illustration of the conformation of Boston Terriers with lower and higher BOAS risk.** (A) Lower risk conformation characteristics: longer muzzle with narrower skull, no abnormal scleral show, open nostrils, and proportionally slimmer neck. (B) Higher risk conformation characteristics: more extreme facial hypoplasia with shortened muzzle and wider skull, abnormal scleral show, stenotic nostrils, and proportionally thicker neck.

Mid-expiratory flow rates have been shown to be useful in this study at demonstrating key differences between BOAS affected and unaffected dogs. The MEF rate changes of Grade 1–3 Boston Terriers show significant differences to the Grade 0 Boston Terriers and control dogs. A large peak in flow at the beginning of the expiratory breath which then rapidly tapers off could be indicative of pharyngeal obstruction. In human patients with hypertrophic soft palate, data collected investigating the biomechanical changes showed a similar peak at the beginning of expiration followed by a reduction with micro pressure sensors and CT scans of the pharynx showing the increase in pressure at the beginning of expiration producing displacement and causing soft palate collapse [53]. Investigation of the internal anatomy of BOAS affected Boston Terriers could aid understanding of the upper airway obstructions and dynamic restrictions that result in changes to the respiratory parameters.

Whilst the main focus of this study was on BOAS in Boston Terriers, there were a number of other findings relating to the brachycephalic and screw-tailed phenotype of the Boston Terrier. Congenital vertebral malformations have been found in screw-tailed breeds [21, 54]. These can sometimes lead to neurological deficits [22]. Some dogs in this study were noted to

have proprioceptive or spinal reflex deficits (n = 6), however without further investigation the cause of these clinical signs remains unknown. In this study, a shorter tail length showed a trend towards increased BOAS risk, however this was not found to be significant. A small number of dogs (n = 2) were reported by owners as having been diagnosed with Chiari-like malformation and syringomyelia (CM/SM); a condition known to affect brachycephalic dogs. CM/SM has been proposed to also be known as brachycephalic obstructive CSF channel syndrome (BOCCS) [55] due to the role brachycephaly plays in disease pathogenesis [56]. Definitive diagnosis requires magnetic resonance imaging (MRI). In terms of ocular disease, this study found that abnormal scleral show is associated with BOAS status. More severely flat-faced dogs are more likely to both have ocular protrusion and BOAS. Scleral show could be considered a sign of brachycephalic ocular syndrome, which can lead to further issues such as a predisposition to corneal ulceration [24]. This study has highlighted several findings relating to brachycephalic conformation-related conditions that may indicate other breed predisposition to disease. Further research may be required to understand the significance of these findings.

## Limitations

Dogs recruited for this study were volunteered by owners that were willing to attend an appointment in Cambridge, UK or participate at shows or health testing days arranged by breeders within the showing community. The ability and motivation to take part may be varied and introduce a sampling bias. For example, an owner that suspects that their dog is affected may be motivated to take part in order to seek a diagnosis or advice regarding treatment. On the other hand, owners that are already aware that their dogs are severely affected may avoid participation due to embarrassment that their dog is suffering from a welfare-compromising condition. Variations may also exist across breed populations of different regions and countries. It remains difficult to determine the true prevalence of BOAS from an unbiased sample population within a breed. Recruitment of a wider population of Boston Terriers across different geographical regions, and longitudinal studies could help to assess the repeatability of these study findings.

Certain factors were limited by the sample size of the study. Conformational factors that showed a trend to significance such as tail length and body condition score may have proven significant with a larger number of dogs. Additionally, the limited sample size of BOAS affected Boston Terriers that underwent whole-body barometric plethysmography (n = 12) meant that attaining quantifiable differences in respiratory parameters between each grade severity (Grade 1 to 3) was not possible. Further work to correlate recorded respiratory noises (stertor and stridor) of increasing severity with the mid-expiratory flow rates in a larger sample of dogs could help to determine this.

## Implications for breeding and clinical practice

The current Boston Terrier breed standard listed on the UK Kennel Club website [57] states that Boston Terriers are to have a muzzle "approximately one-third of length of skull". However, the craniofacial ratio findings demonstrate that most Boston Terriers come up short in this. Breeders and prospective dog owners should exercise caution when choosing which Boston Terriers to breed from or purchase. The more extremely flat-faced Boston Terriers are at greater risk of BOAS, therefore should not be selected for, or bred from. Prospective owners should also check the puppies' parents have been tested for BOAS. When choosing breeding stock, selecting dogs of a more moderate conformation, with open nostrils, no abnormal scleral show and a longer muzzle would be advisable. However, the considerable overlap in

craniofacial ratios between Grade 1–3 and Grade 0 Boston Terriers demonstrates that it is not possible to tell from looks alone which dogs will be affected. Respiratory function grading is an accessible way to determine whether a dog is affected by BOAS and therefore should be utilised in this breed.

When selecting which dogs to breed from, ideally only Grade 0 dogs would be chosen. However, a harsh selection bias for breeding on one factor, BOAS status, in a limited population could result in a genetic bottleneck that carries the risk of unintended propagation of other unrecognised health issues within a breed. This logic has resulted in the current approach by the UK Kennel Club Respiratory Functional Grading Scheme allowing the breeding of Grade 2 dogs with clinically unaffected dogs (Grade 0 and Grade 1). In this sample population, Boston Terriers appear to have a more sizeable pool of Grade 0 dogs with respiratory flow traces that do not differ significantly from control dogs, therefore the prevalence of BOAS within Boston Terriers is lower than in the three more extreme breeds. Incidence of BOAS in Boston Terriers can be reduced further by ensuring that at least one of the parents is Grade 0, and the other is clinically unaffected by BOAS (Grade 0 or 1). Attention also needs to be paid to the general health of the dog, not ignoring other disorders that may impede quality of life and welfare.

It is important for owners and breeders to be vigilant of the clinical signs of BOAS in Boston Terriers and seek veterinary advice and care if concerned. Veterinary surgeons may wish to expand offers of BOAS screening to Boston Terriers so that affected dogs can be treated appropriately. BOAS is a complex syndrome of different upper airway obstructions that can vary across individuals and between breeds. Advanced imaging such as computed tomography and endoscopy are useful to undertake prior to BOAS surgery to identify lesion sites. Further research is underway to identify the internal BOAS lesion sites in Boston Terriers which aims to help identify the most appropriate ways to direct treatment.

## Conclusion

Boston Terriers appear to be less severely affected by BOAS than the most popular extreme brachycephalic screw-tailed breeds. Boston Terriers that are BOAS Grade 0 have similar respiratory traces to control mesocephalic dogs. Whilst there are key conformation differences between brachycephalic breeds that impact BOAS risk, within the Boston Terrier breed more extreme brachycephalic conformation is associated with an increased risk of BOAS. Nostril stenosis, facial foreshortening, abnormal scleral show and a higher NGR are all significantly associated with BOAS status. However, there is considerable overlap between Grade 1–3 and Grade 0 dogs in some measurements, therefore the use of respiratory function grading is more reliable than the detailed conformation for use by owners or breeders in selecting unaffected dogs.

## Supporting information

**S1 Dataset. Boston Terrier BOAS grading, signalment and conformation dataset.**
(XLSX)

**S2 Dataset. Mid-expiratory flow rate WBBP dataset.**
(XLSX)

**S1 Appendix. Repeatability of conformational measurements methodology.**
(DOCX)

**S1 Table. Intraclass correlation coefficient results.** *(*A) Intra-rater repeatability results for one rater across up to 19 subjects of multiple dog breeds in the photographic measurements.

(B) Inter-rater repeatability results for one rater across up to 20 subjects of multiple dog breeds in the photographic measurements. Skull index [1] displaying ICC result for analysis performed on multiple dog breeds, and skull index [2] displaying ICC value for analysis performed on smooth-coated breeds (Boston Terrier, Boxer and Staffordshire Bull Terrier) only. (C) Inter-rater repeatability results for soft tape measurements conducted by two raters across 30 subjects of multiple dog breeds.
(DOCX)

**S2 Table. Data analysis table of mid-expiratory flow (MEF) rate changes during different time periods of the expiratory breath.** Mixed-effects model (REML) and unpaired two-tailed t-tests comparing the differences in change in MEF rate between each sample group. Paired two tailed t-tests comparing MEF rates across different time periods of the expiratory breath within each sample group.
(DOCX)

**S3 Table. Descriptive statistics of conformation and signalment continuous variables.** BOAS Grade 0 Boston terriers and BOAS Grade 1–3 Boston Terriers.
(DOCX)

**S4 Table. Results of the unpaired t-test analysis comparing BOAS unaffected and affected Boston Terriers. Welch's correction applied to variables that do not demonstrate equality of variance (p<0.05 in F test).**
(DOCX)

**S5 Table. Multiple logistic regression full results.**
(DOCX)

**S1 Fig. Bland altman analysis.** *(*A) Plotted craniofacial ratios of the same subjects measured through CT and photographic images. (B) Bland-Altman analysis of CT versus photographic measurements demonstrating risk of bias. CT and photographic measurements of the craniofacial ratio of the same subjects were significantly correlated (p < 0.0001). The left x axis shift of the plotted CT data indicates the underestimation bias of the photographic measurements. Bland-Altman analysis revealed a bias of -0.0499 (SD: 0.0452) for the photographic measurements when compared to the CT measurements.
(TIF)

## Acknowledgments

We thank all the owners and breeders who have brought their dogs to take part in this research, particularly from the Boston Terrier community. Thank you to the staff at the QVSH Cambridge for assistance with running the assessments and recruitment of participants. We give particular thanks to Prof Clare Rusbridge, Dr Mark Reading and Dr Rachael Grundon on their respective expertise for evaluation on clinical examination, Ali Limentani for providing illustrations, and Dr Olivier Restif for advice on statistical analysis on the conformation data.

## Author Contributions

**Conceptualization:** Francesca Tomlinson, Ella O'Neill, Nai-Chieh Liu, David R. Sargan, Jane F. Ladlow.

**Formal analysis:** Francesca Tomlinson, Ella O'Neill.

**Funding acquisition:** Nai-Chieh Liu, Jane F. Ladlow.

**Investigation:** Francesca Tomlinson, Ella O'Neill, David R. Sargan, Jane F. Ladlow.

**Methodology:** Francesca Tomlinson, Ella O'Neill, Nai-Chieh Liu, David R. Sargan, Jane F. Ladlow.

**Supervision:** David R. Sargan, Jane F. Ladlow.

**Validation:** Francesca Tomlinson, David R. Sargan, Jane F. Ladlow.

**Visualization:** Francesca Tomlinson, Nai-Chieh Liu.

**Writing – original draft:** Francesca Tomlinson.

**Writing – review & editing:** Francesca Tomlinson, Ella O'Neill, Nai-Chieh Liu, David R. Sargan, Jane F. Ladlow.

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
