## [Decision Letter · Decision Letter 0]

7 Jun 2024

PONE-D-24-10830BOAS in the Boston Terrier: a healthier screw-tailed breed?PLOS ONE

Dear Dr. Ladlow,

Thank you for submitting your manuscript to PLOS ONE. After careful consideration, we feel that it has merit but does not fully meet PLOS ONE’s publication criteria as it currently stands. Therefore, we invite you to submit a revised version of the manuscript that addresses the points raised during the review process.

We look forward to receiving your revised manuscript.

Kind regards,

Ioannis Savvas, DVM, Ph.D.

Academic Editor

PLOS ONE

Journal Requirements:

Reviewers' comments:

Reviewer's Responses to Questions

**Comments to the Author**

1. Is the manuscript technically sound, and do the data support the conclusions?

Reviewer #1: Yes

Reviewer #2: Yes

2. Has the statistical analysis been performed appropriately and rigorously? 

Reviewer #1: I Don't Know

Reviewer #2: Yes

3. Have the authors made all data underlying the findings in their manuscript fully available?

Reviewer #1: Yes

Reviewer #2: Yes

4. Is the manuscript presented in an intelligible fashion and written in standard English?

Reviewer #1: Yes

Reviewer #2: Yes

5. Review Comments to the Author

Reviewer #1: The authors effectively provide background information on the significance of brachycephalic obstructive airway syndrome (BOAS) in certain dog breeds, emphasizing related welfare concerns. They introduce their research question, aiming to investigate the prevalence and risk factors for BOAS in Boston Terriers, a breed known for its brachycephalic features. The study, involving 104 Boston Terriers, comprehensively assesses respiratory function and conformational characteristics. By comparing Boston Terriers to other brachycephalic breeds and delving into genetic and health similarities, the authors contextualize their research within the broader field of brachycephaly-related health issues. Overall, this study provides valuable insights into the prevalence and risk factors associated with BOAS in Boston Terriers.

Although not explicitely stated, the authors suggest some key findings, like how their study might help breeders and owners choose Boston Terriers less likely to have BOAS.

The authors also mention other research on BOAS in popular brachycephalic breeds and efforts to address welfare concerns. While they don't go into detail about how their study connects to this research, they imply that their work will help fill gaps in our knowledge about BOAS in Boston Terriers compared to other breeds, which could be valuable for future research.

While the methodology, including whole-body barometric plethysmography and clinical examination, is sound. However, limitations related to sampling bias due to volunteer-based recruitment are acknowledged. Authors might want to consider discussing how this bias could affect how well the findings apply to other groups and suggest ways to fix it, like getting participants from more places (multi-center recruitment).

There are also sample size limitations in analyzing differences between BOAS severity grades. Please address the limitations imposed by the sample size, particularly in analyzing differences between BOAS severity grades. Addressing these limitations, discussing potential implications for the interpretation of the results and consider ing suggestions for future research to address this limitation would strenghten the study’s conclusions.

Clear presentation of results and interpretation of findings in the context of previous research and breed standards is given in the discussion.

Plethysmography serves as a vital tool for evaluating respiratory function, especially in studies investigating brachycephalic obstructive airway syndrome (BOAS) in dogs. However, the study's efficacy may be affected by its sample size. With only 12 BOAS-affected Boston Terriers undergoing whole-body barometric plethysmography, the sample size is relatively small, potentially limiting the statistical power and confidence in the findings regarding respiratory parameters. Moreover, this limitation may have hindered the observation of quantifiable differences in respiratory parameters across different BOAS severity grades. Therefore, while plethysmography is valuable, the study's small sample size should be considered when interpreting the results. Future research with a larger sample size would enhance the validation and robustness of findings concerning respiratory function in BOAS-affected Boston Terriers.

The findings of this study are likely applicable to Boston Terrier populations in similar settings, although variations may exist across different regions. However, it's important to acknowledge that other researchers may have different experiences. Factors such as geographical location and breeding practices could influence the health status of Boston Terriers. For instance, anecdotal evidence suggests that populations in Southern Europe may exhibit higher rates of severe BOAS cases compared to those in other regions. Collaborative research efforts across different regions and populations of Boston Terriers may provide a more comprehensive understanding of BOAS and its associated risk factors

While the main focus is on BOAS, the study also identifies other health issues related to the brachycephalic and screw-tailed phenotype of Boston Terriers, including congenital vertebral malformations, neurological deficits, and ocular abnormalities. However, further research is needed to fully understand the significance of these findings and their implications for breed predisposition to disease.

Overall, the discussion provides a comprehensive overview of the study findings, highlighting the importance of understanding BOAS and its associated risk factors in Boston Terriers, while also acknowledging the need for further research to address other health issues in this breed.

I recommend acceptance for publication with minor revisions to address the identified limitations and clarify certain points.

Reviewer #2: Comments on specific lines:

Line 250 - I was uncertain why a 1-sided test was chosen for analysing the proportion of Grade 0 dogs? Wouldn't it be equally of interest if % Grade 0s in Boston Terriers was either higher or lower than the sample population?

Line 256 - the time periods for the flow rate analysis are referred to sometimes as percentages of the total time and sometimes as decimal fractions. It would be better to keep this consistent throughout.

Line 257 - I'm wondering why you didn't make use of your mixed model to compare the flow rate changes? It would be more usual, I think, to use your model coefficients and their variance-covariance matrix to calculate the predicted flow rate change differences and their variance, and test that way.

Line 261-2 - Is there a word missing in this sentence somewhere? It didn't quite make sense to me.

Line 272 - Again I'm wondering why the choice to use a 1-sided test for the continuous BOAS risk factors.

Line 273 - Should the list of continuous variables in brackets also include EWR, SI and CFR?

Line 339 - I'm not sure you'd really call the change in flow rate in the grade 0s 'steady' as this implies a pretty constant rate - it looks like there is a rather lower rate of increase for 0.5-0.75 than in 0.25-0.5. 'Steady' is much more applicable to the control dogs, though.

Lines 341-2 - I found this sentence to be a little unclear. I think it would help to clarify that you aren't talking about all the groups here just the control and grade 0 dogs.

Line 365 - Whilst the neck girth ratio is clearly significant, 0.62 vs 0.63 seems like a very small difference. Is this small difference likely to be practically / clinically significant?

General comments:

Overall, I think this is an interesting paper on an important subject, with a wide variety of variables looked at, and contributes useful information to those interested in issues around brachy breeds.

6. PLOS authors have the option to publish the peer review history of their article (what does this mean?). If published, this will include your full peer review and any attached files.

Reviewer #1: **Yes: **Vladimira Erjavec

Reviewer #2: No

---

## [Author Response · Author response to Decision Letter 0]

22 Jul 2024

Dear Ioannis Savvas,

Many thanks for your response to our manuscript submission. We thank the reviewers for their comments and feedback.

The manuscript and reference list have been reviewed to ensure they meet PLOS ONE’s style requirements. We attach the revised manuscript and below we address the points raised by the reviewers.

We hope that the revised manuscript will now be suitable for publication and look forward to hearing from you. 

Kind regards,

…………………..

Reviewer 1

In response to Reviewer #1’s comments, additional remarks regarding the limitations of the study and potential ways to address this have been added on lines 526-530.

Reviewer 2

The following answers are made in response to Reviewer #2’s comments organised by line originally referenced:

Line 250 - A one-sided test was chosen as it was hypothesised that Boston Terriers had a higher percentage of Grade 0 Boston Terriers (lines 97-98).

Line 256 - Amended lines 164-165 to ensure consistent language used throughout text.

Line 257 - Repeated measures ANOVA with a mixed model approach was used to look at the temporal trends between the three groups (control, Grade 0 and Grade 1-3). Using Prism, I don’t believe that would be possible with this type of analysis. The mixed effects model used accounts for the subjects being random effects to enable analysis of temporal trends, whilst each subject can start from a different flow rate level. Pairwise comparisons were therefore then used to look at the data in more granular detail. The unpaired t-tests were performed for comparing the flow rate change between groups, and paired t-tests for flow rate change within groups.

Line 261-2 - Amended sentence.

Line 272 - A one-sided test was used for continuous risk factors as it was hypothesised that more ‘extreme’ conformation was associated with BOAS, and is more sensitive at detecting a smaller effect size. 

Line 273 - EWR, SI and CFR are listed on line 276.

Line 339 - Amended to ‘smaller change’ on line 340.

Lines 341-2 - Amended on line 343 to clarify that talking about the two groups (control and Grade 0).

Line 365 - Whilst it is likely difficult to differentiate on a practical level the difference between a Boston Terrier with a neck girth ratio of 0.62 compared to 0.63, noting this as a significant factor could help to identify dogs that have a more extreme thicker neck compared to chest girth as potentially being at risk of BOAS (the maximum NGR Grade 1-3 was 0.76 compared to 0.68 for Grade 0s).

---

## [Decision Letter · Decision Letter 1]

30 Sep 2024

PONE-D-24-10830R1BOAS in the Boston Terrier: a healthier screw-tailed breed?PLOS ONE

Dear Dr. Ladlow,

Thank you for submitting your revised manuscript to PLOS ONE. Please amend the manuscript according to a minor comment by one of the reviewers, before it can be accepted for publication.

We look forward to receiving your revised manuscript.

Kind regards,

Ioannis Savvas, DVM, Ph.D.

Academic Editor

PLOS ONE

Journal Requirements:

Reviewers' comments:

Reviewer's Responses to Questions

**Comments to the Author**

1. If the authors have adequately addressed your comments raised in a previous round of review and you feel that this manuscript is now acceptable for publication, you may indicate that here to bypass the “Comments to the Author” section, enter your conflict of interest statement in the “Confidential to Editor” section, and submit your "Accept" recommendation.

Reviewer #1: All comments have been addressed

Reviewer #2: (No Response)

2. Is the manuscript technically sound, and do the data support the conclusions?

Reviewer #1: Yes

Reviewer #2: Yes

3. Has the statistical analysis been performed appropriately and rigorously? 

Reviewer #1: I Don't Know

Reviewer #2: Yes

4. Have the authors made all data underlying the findings in their manuscript fully available?

Reviewer #1: Yes

Reviewer #2: Yes

5. Is the manuscript presented in an intelligible fashion and written in standard English?

Reviewer #1: Yes

Reviewer #2: Yes

6. Review Comments to the Author

Reviewer #1: The manuscript presents a well-organized and thorough study on BOAS in Boston Terriers. The manuscript is well-organized, and the data support the conclusions. The methods used are suitable for the study's goals. The statistical analysis seems appropriate, with clear presentation of results, though I am not an expert in statistics and therefore cannot fully assess the appropriateness of the statistical methods used.

The authors have complied with data availability requirements, making the underlying data easily accessible. The manuscript is clearly written, making it straightforward to follow. The discussion is particularly valuable, especially in addressing the differences between Boston Terriers and other brachycephalic breeds, adding useful insights to the existing knowledge.

Reviewer #2: Thank you for addressing the majority of my previous comments. There is one that is still outstanding though - the sentence on line 261-2 that I found to be unclear was this one - 'In the absence of missing values, P values and multiple comparison tests are given the same as repeated measures ANOVA'. In the revised manuscript, it looks like you altered the sentence before this one, which was already fine! I think that it should probably(?) say something like 'In the absence of missing values, P values and multiple comparison tests given by this method are the same as would be obtained by repeated measures ANOVA'. Apologies for the confusion over which sentence I was talking about in my original comments!

7. PLOS authors have the option to publish the peer review history of their article (what does this mean?). If published, this will include your full peer review and any attached files.

Reviewer #1: No

Reviewer #2: No

---

## [Author Response · Author response to Decision Letter 1]

24 Nov 2024

Many thanks for the thorough review of this paper. I have uploaded an amended manuscript with the outstanding change amended for reviewer 2,

Many thanks

Jane Ladlow

---

## [Decision Letter · Decision Letter 2]

26 Nov 2024

BOAS in the Boston Terrier: a healthier screw-tailed breed?

PONE-D-24-10830R2

Dear Dr. Ladlow,

We’re pleased to inform you that your manuscript has been judged scientifically suitable for publication and will be formally accepted for publication once it meets all outstanding technical requirements.

Kind regards,

Ioannis Savvas, DVM, Ph.D.

Academic Editor

PLOS ONE

Additional Editor Comments (optional):

Reviewers' comments:

Reviewer's Responses to Questions

**Comments to the Author**

1. If the authors have adequately addressed your comments raised in a previous round of review and you feel that this manuscript is now acceptable for publication, you may indicate that here to bypass the “Comments to the Author” section, enter your conflict of interest statement in the “Confidential to Editor” section, and submit your "Accept" recommendation.

Reviewer #2: All comments have been addressed

2. Is the manuscript technically sound, and do the data support the conclusions?

Reviewer #2: Yes

3. Has the statistical analysis been performed appropriately and rigorously? 

Reviewer #2: Yes

4. Have the authors made all data underlying the findings in their manuscript fully available?

Reviewer #2: Yes

5. Is the manuscript presented in an intelligible fashion and written in standard English?

Reviewer #2: Yes

6. Review Comments to the Author

Reviewer #2: (No Response)

7. PLOS authors have the option to publish the peer review history of their article (what does this mean?). If published, this will include your full peer review and any attached files.

Reviewer #2: No

---

## [Editor Report · Acceptance letter]

9 Dec 2024

PONE-D-24-10830R2 

PLOS ONE

Dear Dr. Ladlow, 

I'm pleased to inform you that your manuscript has been deemed suitable for publication in PLOS ONE. Congratulations! Your manuscript is now being handed over to our production team.

Kind regards, 

on behalf of

Prof. Ioannis Savvas 

Academic Editor

PLOS ONE